# Evaluating Levels of Community Participation in a University-Community Partnership: The Jackson Heart Study

**DOI:** 10.3390/diseases10040068

**Published:** 2022-09-23

**Authors:** Clifton Addison, Brenda W. Campbell Jenkins, Marty Fortenberry, Darcel Thigpen-Odom, Pamela McCoy, Lavon Young, Monique White, Gregory Wilson, Clevette Woodberry, Katherine Herron, Donna Antoine LaVigne

**Affiliations:** Jackson Heart Study, Jackson State University, Jackson, MS 39213, USA

**Keywords:** community outreach and engagement, CBPR, Jackson Heart Study, African Americans

## Abstract

Objective: This research was designed to evaluate the perceptions of the Jackson Heart Study (JHS) community relating to their levels of involvement in JHS activities that were developed to address health disparities and promote health education and health promotion. Methods: The participants for this study comprised 128 community members, who included JHS participants, as well as family members and other friends of the JHS who resided in the JHS community of Hinds, Madison, and Rankin Counties in Mississippi and attended the JHS Annual Celebration of Life. We used the Chi-Square test to analyze the participants’ responses to the survey questions developed to address the six areas of focus: (1) ways to increase participation in community outreach activities; (2) reasons for participating in community outreach activities; (3) interest in research participation; (4) factors influencing engagement; (5) Participants’ preferences for communicating; (6) Chronic disease prevalence. Results: Participants residing in rural counties perceived television and radio as a medium to increase participation; More female respondents cited trust working with the JSU JHS Community Outreach Center (CORC) as a reason for remaining engaged in the community outreach activities; younger participants under 66 years of age recommended social media as a way to increase participation; participants residing in the rural areas saw their participation in the community outreach activities as a way to address community health problems. Conclusions: The knowledge gained from the details provided by the JHS community members can be used to refine research studies in existence, while promoting their sustainability.

## 1. Introduction

It is widely accepted that disadvantaged groups are exposed to a greater degree of health risk factors and are affected by worse health outcomes, such as prevalence of chronic diseases, like cardiovascular diseases (CVD), compared to non-disadvantaged groups [1,2,3]. It has become evident to stakeholders interested in addressing health disparities, risk factors, and health status in minority, racial, and ethnic populations that community-based participatory research can contribute to the development of important policy initiatives that can designate individuals or organizations with specific authority and responsibility for reducing health disparities in their communities [4,5]. This type of community participation is beneficial to the development, implementation and evaluation of health services [6] and the implementation of culturally customized interventions to counteract the complex interaction of biological, sociopolitical, economic, and environmental factors that contribute to the disparities that exist in these communities [7].

The public health burden caused by cardiovascular disease (CVD) continues to adversely affect individuals through cost, life expectancy, medical, pharmaceutical and hospital care, particularly in the case of African Americans [8,9,10,11]. The Jackson Heart Study (JHS) Community Outreach Center (CORC) was developed by Jackson State University (JSU) to build a collaborative health promotion partnership with the objective of effectively employing principles of community-based participatory research (CBPR) in the process of promoting academic-community interactions and partnerships to address issues relating to health disparities. The resulting JSU CORC CBPR platform involved many dimensions of collaboration. These dimensions of collaboration sought to ensure that the partnerships that were configured included groups of diverse community representatives who were guided by a vision for long-term change so that positive results derived from their activities and interactions would be long-lasting [12].

The value of shared academic-community decision-making and research study collaboration has become increasingly evident in research operations as community engagement practices have become more valued [13]. Green and Haines (2012) acknowledged that the community voice and perspective were not being adequately utilized. However, the research on community engagement reveals a lack of understanding of the impact of community voice and perspective on the operations and efficiency of these academia-community partnerships [14]. Cruz and Giles (2000) suggested that it is important for stakeholders engaged in an academic-community collaboration to be able to evaluate how well the arrangement if progressing because of the quality of the partnership [15]. To facilitate harmonization of its community mobilization agenda, the JSU JHS CORC sponsored three annual community events-Celebration of Life, Community Monitoring Board Meeting; Birthday Celebration [16,17]. This research was designed to evaluate the perceptions of the JHS community relating to their levels of involvement in JHS activities that were developed to address health disparities and promote health education and health promotion.

## 2. Materials and Methods

The JHS, the largest single-site epidemiological study of cardiovascular disease in African Americans, collected data from 5306 African Americans in Hinds, Madison and Rankin Counties (Mississippi). The Jackson Heart Study is a community-based cohort study evaluating the etiology of cardiovascular, renal, and respiratory diseases among African Americans residing in the three counties (Hinds, Madison, and Rankin) that make up the Jackson, Mississippi metropolitan area. Data and biologic materials have been collected from 5306 participants. The age at enrollment for the cohort was 35–84 years. Participants provided extensive medical and social history and had an array of physical and biochemical measurements and diagnostic procedures during a baseline examination (2000–2004) and two follow-up examinations (2005–2008 and 2009–2012). Genomic DNA was collected during the first two examinations. Annual follow-up interviews and cohort surveillance are ongoing, and the fourth examination began in 2022.

The JHS and all of its centers were funded through contracts with the National Heart, Lung, and Blood Institute (NHLBI), National Institutes of Health (NIH). Details of the JHS, its development, participants, and overall procedures are described in earlier publications [18,19,20,21]. The JSU JHS community initiatives included an innovative community-driven operation that devised techniques to motivate, inspire, and engage community residents in a community–academia partnership to yield maximum benefits in the areas of health education, health promotion and interventions, and biomedical research [16].

The data were collected at one of the annual community events sponsored by the JHS Community Outreach Center (CORC) in Jackson, Mississippi. This was one of the largest JHS educational and retention events, and was held in the month of February in commemoration of African American History Month and American Heart Month. The meeting was attended by about 140 JHS participants, community members and other stakeholders, such as JHS community health advisors (CHAs), members of the JHS community ethics advisory board (CEAB), ministers, and partners in community-based and faith-based organizations [17]. All of the attendees at the event were invited to participate in the survey. Event attendees were provided the survey at the registration table when signing in for the event. All respondents were notified of the study’s purpose and given an informed consent form prior to data collection. The completed surveys were placed in a sealed envelope to ensure anonymity in the process. We analyzed data from 128 of the participants who provided completed surveys. The participants’ responses to each area of interest on the survey are provided in the Chi-Square tables in the results section with a tabulation of the frequency of selection of each of the response choices among the 128 participants (enumerating the number of participants who selected each reponse choice).

Survey items relating to the JHS activities were developed from results of past JHS focus groups and other prior JHS community activities, and the validity of the content was assessed. The scale construction was finalized after pre-testing the questions, administering the survey, reducing the number of items, and determining the number of factors. Reliability was tested using Cronbach Alpha (0.78). and validity was assessed. The survey represented an opportunity for the JHS community to express their opinions and concerns about the JHS and provide ideas and input for the future direction of the study. For the purposes of this paper, only the participant recruitment and engagement-related items were explored.

All methods were performed in accordance with the relevant ethical guidelines and regulations. Ethical approval for this study was obtained from the Institutional Review Board (IRB) at Jackson State University, Jackson, Mississippi, USA. Participants were advised that their participation in the survey was voluntary, and that they could stop anytime they wanted, or not answer if they were uncomfortable with individual questions. Participants in this research provided signed informed consent prior to the beginning of the research activities.

We collected data from the participants to address the following six areas that highlighted the participants’ perceptions regarding their level of engagement: (1) Exploring ways to increase participation in community outreach activities; (2) Exploring reasons for participating in community outreach activities; (3) Exploring participants’ interest in research participation; (4) Factors influencing participants’ continuing engagement in community engagement activities; (5) Participants’ preferences for involving and communicating about community engagement activities; (6) Chronic disease prevalence by individual counties/communities. The perceptions of the community members were obtained from the surveys that were administered at the Celebration of Life, one of the JHS annual community events. Community participants were asked to provide answers to the following questions:What are some ways we can increase participation in the community outreach activities?To what extent did the following reasons cause you to first participate in the community outreach activities?Would you be interested in participating in a community research study similar to the Jackson Heart Study?How important are the following items in keeping you engaged in the community outreach activities?What is the best way to communicate to you about community outreach activities?What diseases affect the community where you live the most?

We used quantitative measures to examine the participants’ perceptions that were gathered at this annual event. The participants were asked to select their level of agreement/disagreement to each question posed on the survey by selecting from the four choices provided. The Chi-Square test was used to analyze the participants’ responses to the questions developed to address the six areas of focus. Significance was set at alpha = 0.05.

## 3. Results

To maximize CORC’s efforts to increase community participation in research study activities, our researchers focused specifically on participants’ responses where significant differences were noted. We believed that a review of the differences in the participants’ responses would serve to identify areas where additional attention and efforts may be needed in further interactions with community partners to boost their willingness to participate. Situations where significant differences were noted are discussed below.**Participant Characteristics**

As shown in Table 1, a total of 128 individuals completed the questionnaire. Of those, 76.4% (*n* = 94) were female and 96.8% (*n* = 122) self-identified as Black or AA. Among all respondents, 42.4% (*n* = 53) were sixty-six years of age or older. Furthermore, 16.8% (*n* = 21) were 56–65 years of age, 18.4% (*n* = 23) noted 46–55 years of age, 10.4% (*n* = 13) selected the 36–45 years of age group; and 35 years of age or younger was identified by 12% (*n* = 15) of participants. Regarding county of residence, 68.5% (*n* = 87) of participants lived in Hinds, 16.5% (*n* = 21) resided in Madison and nine respondents (7.1%) were residents of Rankin county. Approximately eight percent (*n* = 10) lived outside of the tri-county region. Participants were then asked about their CORC affiliation, with multiple options to select. Nearly thirty-four percent (*n* = 41) were community members, 30.6% (*n* = 37) were current JHS participants and 22.3% (*n* = 27) were students within an academic institution. Staff, institutional partners and representatives of non-profits combined constituted 13.2% (*n* = 16) of those who completed the questionnaire.

### 3.1. Exploring Ways to Increase Participation in Community Outreach Activities

Community participants were asked: What are some ways we can increase participation in the community outreach activities?

#### 3.1.1. Using Television/Radio by County of Residence

The perceptions of residents in Hinds County were compared to the combined perceptions of residents from Madison and Rankin Counties, the rural communities. While more than half of the total number of participants suggested the use of television and radio, respondents from the combined counties, which were more rural than Hinds County perceived television and radio as a way to increase participation at a significantly higher rate than respondents from Hinds County (Table 2).

#### 3.1.2. Using Social Media by Age

Younger participants under 66 years of age recommended social media as a way to increase participation at a significantly higher rate than older respondents, those above 66 years old (Table 3).

### 3.2. Exploring Reasons for Participating in Community Outreach Activities

Community participants were asked: To what extent did the following reasons cause you to first participate in the community outreach activities?

#### 3.2.1. Addressing Health Problems by County

More participants living in the combined counties of Madison and Pearl Counties, the rural areas of the study area saw their participation in the community outreach activities as a way to address health problems in their community (Table 4).

#### 3.2.2. Improving Personal Heath and Quality of Life by Sex

More females reported improving personal health and quality of life as a reason for choosing to participate in the community outreach activities compared to men (Table 5).

### 3.3. Exploring Participants’ Interest in Research Participation

Community participants were asked: Would you be interested in participating in a community research study similar to the Jackson Heart Study?

#### 3.3.1. Research on Physical Activity by Age

Younger respondents (ages 19–65) reported being more interested in participating in a research study focusing on physical activity when compared to older respondents (Table 6).

#### 3.3.2. Research Study on Asthma by Age

The oldest respondents (ages 66 and older) reported being more interested in participating in a research study focusing on asthma when compared to younger respondents (Table 7).

#### 3.3.3. Research Study on Mental Health by Age

Younger respondents (ages 19–45) reported being more interested in participating in a research study focusing on mental health when compared to older respondents (Table 8).

### 3.4. Factors Influencing Participants’ Continuing Engagement in Community Engagement Activities

Community participants were asked: How important are the following items in keeping you engaged in the community outreach activities?

#### 3.4.1. Importance of Trust by Age

Older respondents (ages 46 and above) reported their trust in the staff working with the JSU CORC as a reason for remaining engaged in the community outreach activities (Table 9).

#### 3.4.2. Importance of Trust by Sex

More female respondents cited trust working with the JSU JHS CORC as a reason for remaining engaged in the community outreach activities (Table 10).

### 3.5. Participants’ Preferences for Communication about Community Engagement Activities

Community participants were asked: What is the best way to communicate to you about community outreach activities?

#### 3.5.1. Email by County of Residence

Respondents from the two combined rural counties (Madison and Rankin) reported that email is the best way to communicate to respondents about community outreach activities at a significantly higher rate than respondents from Hinds County (Table 11).

#### 3.5.2. Text Messages by County of Residence

Respondents from the combined rural counties reported text messages as the best way to communicate to respondents about community outreach activities at a significantly higher rate than respondents from Hinds County (Table 12).

#### 3.5.3. Mail Reminders by Sex

Females reported mail reminders as the best way to communicate to respondents about community outreach activities at a significantly higher rate than males (Table 13).

#### 3.5.4. Email by Age

Younger respondents reported email as the best way to communicate to respondents when compared to older respondents (Table 14).

#### 3.5.5. Telephone Follow-Up Calls by Age

Older respondents reported telephone follow-up calls as the best way to communicate to respondents about community outreach activities when compared to younger respondents (Table 15).

#### 3.5.6. Text Messages by Age

Younger respondents reported text messages as the best way to communicate to respondents about community outreach activities at a significantly higher rate than older respondents (Table 16).

#### 3.5.7. Mail Reminders by Age

Older respondents reported mail reminders as the best way to communicate to respondents about community outreach activities when compared to younger respondents (Table 17).

### 3.6. Chronic Disease Prevalence by Individual Counties/Communities

Community participants were asked: What diseases affect the community where you live the most?

#### 3.6.1. Heart Disease by County

Respondents from both the other combined counties and Hinds County reported that heart disease significantly affects their community (Table 18).

#### 3.6.2. High Blood Pressure by Sex

More males than female respondents reported perceived that high blood pressure disease significantly affects their community (Table 19).

## 4. Discussion

As researchers explore ways to increase participation in community outreach activities, capturing the attention of the community can be a difficult task. When engaging African Americans to participate in research and research activity, recruiters and other community advocates represent the first point of contact that can motivate the individuals to make the commitment to participate [22]. The results of the analyses of the participants’ perceptions can serve as important tips for future researchers who aspire to recruit community members, particularly African Americans and other minorities, to participate in research studies. The results of this research provide strategies to drive community engagement, and tips on how to promote public participation.**Limitations and Strengths**

This study has both limitations and strengths. Using a convenience sampling strategy in selecting the study participants is a limitation because some important JHS participants and community partners might have been missed. The small number of participants is a limitation. Some bias may be present because the research was conducted at a JHS community event. As a result, participants would have had a heightened awareness of the communication being administered to them because they were themselves, discussing how to effectively disseminate messages to their community audiences.

The design of this study provided numerous benefits. The choice to administer the survey to a group that included JHS cohort members, family members, local community, and other stakeholders around the Jackson Metropolitan Area, was also a benefit as it meant data were based on a variety of first-hand experiences from individuals with an active interest in promoting JHS activities. The results are therefore applicable and meaningful to CBPR research. The limitations mentioned suggest that our findings might not be easily generalizable to the population. Therefore, additional studies are encouraged to investigate whether different populations’ perceptions of appropriate communication strategies are different from our findings. The lessons learned from this study should be received as meaningful resources when developing strategies to inform the development of specific health information dissemination solutions.

Below are listed important tips and strategies that can be employed to increase and broaden community engagement:

### 4.1. Exploring Ways to Increase Participation in Community Outreach Activities

The JHS community participants perceived television, radio and social media as a way to increase participation at a significantly higher rate. This would make it possible to define a range of participant and community engaged research activities that are understood and accepted by both the academic researchers and collaborating community members [23]. This type of collaborative approach is important for building trust and understanding barriers to community participation in research activities [7].

### 4.2. Exploring Reasons for Participating in Community Outreach Activities

Understanding community members’ reasons for participating in community outreach activities and why it is important for community members to participate in community outreach activities can assist researchers to be successful in recruiting African Americans and minority participants in research studies. Community outreach activities increases social consciousness and social awareness and responsibility as well. By committing to a project or activity with others, participation by community members helps to build and strengthen relationships and make new friendships. The participants believed that it serves as a way to address health problems in their community, as well as improving personal health and quality of life. If designed and implemented properly through effective community consultation and participation, these types of community engagement collaborations can result in improved health and health behaviors, especially among disadvantaged populations [24].

### 4.3. Exploring Participants’ Interest in Research Participation

It is also important to exploring participants’ interest in research participation. The need to keep research organization staff connected and engaged is very important. Understanding the participants’ perceptions can assist public health investigators in transforming participants’ continuing engagement in research. The research process was able to gather ideas from the community, analyzed their input and reported the accounts so that future researchers can make better-informed decisions. Participants’ perceptions reflected their focus on risk factors like physical activity and on factors like asthma. Long et al. (2016) suggested that studies should begin to shift their approach to involve participants’ preferences [25].

### 4.4. Influencing Participants’ Continuing Engagement in Community Engagement Activities

Factors influencing participants’ continuing commitment to community engagement activities—The need to keep study participants and community members connected and engaged is very important to the success of any research enterprise. Participants’ reported their trust in the staff working with the JSU CORC as a major reason for remaining engaged in the community outreach activities. According to Deweger et al. (2018), Effective participation of community members is only possible when processes are included that the research organizations are inclusive, accessible, and supportive of community members [26].

### 4.5. Participants’ Preferences for Communication about Community Engagement Activities

Participants’ preferences for involving and communicating about community engagement activities are important. In order to ensure useful interacting and effectively communicating with community members throughout the research process. The ability to communicate effectively and place the community at the center of the response spectrum is crucial to successful recruitment and retention. Participants’ perceptions regarding preferences for communication included mail, email and text messages as the best way to communicate to them about community outreach activities. Expressions of a reciprocal relationship over time serve as a catalyst to developing important levels of trust. These levels of engagement can begin with adherence to ethical conduct and communication to receiving gifts for their participation [27].

## 5. Conclusions

The knowledge gained from the details provided by the JHS community members can be used to refine research studies in existence, while promoting their sustainability. The successful implementation of effective community engagement strategies can expand the avenues available for employing community-based strategies that can facilitate health improvement in the communities. Effective community engagement can address socioeconomic issues that are the root causes of many health issues. By engaging with partners in the community to address socioeconomic issues, health care organizations can assist in improving health outcomes, thereby reducing the costs of services. Understanding realistic health issues that affect community health, implementing effective policies to address public health, and developing meaningful prevention and intervention strategies to reduce prevalence of chronic diseases can best be achieved when effective community engagement programs are appreciated, installed, and operating. The extent of the collaboration between participating stakeholders will help to ensure that the community benefits from the wealth of perspectives, insight, and input to the maximum extent possible. A community building framework with community events like the JHS CORC events can be a mechanism through which information and insights can be gathered that can facilitate practitioners work toward communication and information dissemination goals.

## Figures and Tables

**Table 1 diseases-10-00068-t001:** Demographic Characteristics of Participants.

Characteristic	Respondents (*n* = 128) % (*n*)
Gender	
Female	76.4 (94)
Male	23.6 (29)
Race	
African American (Black)	96.8 (122)
Caucasian (White)	3.2 (4)
Age	
35 years or younger	12.0 (12)
36–45	10.4 (13)
46–55	18.4 (23)
56–65	16.8 (21)
66 year or older	42.4 (53)
County of Residence	
Hinds	68.5 (87)
Madison	16.5 (21)
Rankin	7.1 (9)
Other	7.9 (10)
Affiliation	
Community Member	33.9 (41)
JHS Participant	30.6 (37)
Community Health Advisor	13.2 (16)
Staff	5.8 (7)
Institutional Partners	2.5 (3)
Non-Profit	4.9 (6)
Student	22.3 (27)
Other	1.7 (2)
**Years Active as CORC Participant**	
0–1 year	44.4 (56)
2–4 years	10.3 (13)
5–8 years	4.8 (6)
9–11 years	5.6 (7)
12 years of more	34.9 (44)
Role in CORC’s Activities	
Planning community activities	28.7 (29)
Outreach for recruitment	23.8 (24)
Hosting community health fairs	21.8 (22)
Health education messages	19.8 (20)

**Table 2 diseases-10-00068-t002:** Using television/radio to increase participation in community outreach activities by county of residence.

	County			Total
		Hinds	Other Combined	
No Television/Radio	Count	43	12	55
	Expected Count	37.4	17.6	55.0
	% Within County	49.4%	29.3%	43%
Television/Radio	Count	44	29	73
	Expected Count	49.6	23.4	73
	% Within County	50.6%	70.7%	57.0%
Total	Count	87	41	128
	Expected Count	87	41	128.0
	% Within County	100.0%	100.0	100.0%

*χ*^2^ = 4.620, *df* = 1, *p* = 0.032.

**Table 3 diseases-10-00068-t003:** Cross-tabulation results for using social media to increase participation in community outreach activities by age.

	Age					Total
		Up to 18	19–45	46–65	66 and Above	
No Social Media	Count	1	6	13	41	61
	Expected Count	1.0	12.4	21.0	26.7	61.0
	% Within Age	50.0%	23.1%	29.5%	73.2%	47.7%
Social Media	Count	1	20	31	15	67
	Expected Count	1.0	13.6	23.0	29.3	67.0
	% Within Age	50.0%	76.9%	70.5%	26.8%	52.3%
Total	Count	2	26	44	56	128
	Expected Count	2.0	26.0	44.0	56.0	128.0
	% Within Age	100.0%	100.0%	100.0%	100.0%	100.0%

*χ*^2^ = 26.751, *df* = 1, *p* = 0.000.

**Table 4 diseases-10-00068-t004:** Addressing health problems in the community by county of residence.

	County			Total
		Hinds	Other Combined	
None	Count	0	4	4
	Expected Count	2.6	1.4	4.0
	% Within County	0%	10%	3.5%
Slightly	Count	1	5	6
	Expected Count	2.1	3.9	6.0
	% Within County	2.5%	6.8%	5.3%
Moderately	Count	9	8	17
	Expected Count	6.0	11.0	17.0
	% Within County	22.5%	10.8%	14.9%
Very	Count	9	27	36
	Expected Count	12.6	23.4	36.0
	% Within County	22.5%	36.5%	31.6%
Extremely	Count	17	34	51
	Expected Count	17.9	33.1	51.0
	% Within County	42.5%	45.9%	44.7%
Total	Count	40	74	114
	Expected Count	40.0	74.0	114.0
	% Within County	100.0%	100.0%	100.0%

*χ*^2^ = 12.350, *df* = 4, *p* = 0.015.

**Table 5 diseases-10-00068-t005:** Cross-tabulation results for improving personal health and quality of life by sex.

	Sex			Total
		Male	Female	
None	Count	3	0	3
	Expected Count	0.7	2.3	3.0
	% Within Sex	11.1%	0%	2.6%
Slightly	Count	1	1	2
	Expected Count	0.5	1.5	2.0
	% Within Sex	3.7%	1.1%	1.8%
Moderately	Count	3	8	11
	Expected Count	2.6	8.4	11.0
	% Within Sex	11.1%	9.2%	9.6%
Very	Count	6	20	26
	Expected Count	6.2	19.8	26.0
	% Within Sex	22.2%	23.0%	22.8%
Extremely	Count	14	58	72
	Expected Count	17.1	54.9	72.0
	% Within Sex	51.9%	66.7%	63.2%
Total	Count	27	87	114
	Expected Count	27.0	87.0	114.0
	% Within Sex	100.0%	100.0%	100.0%

*χ*^2^ = 11.233, *df* = 4, *p* = 0.024.

**Table 6 diseases-10-00068-t006:** Cross-tabulation results for research study focusing on physical activity by age.

	Age					Total
		Up to 18	19–45	46–65	66 and Above	
Interested in participating in research focusing on physical activity	Count	2	22	34	27	85
	Expected Count	1.3	17.3	29.2	37.2	85.0
	% Within Age	100.0%	84.6%	77.3%	48.2%	66.4%
Not interested in participating in research focusing on physical activity	Count	0	4	10	29	43
	Expected Count	0.7	8.7	14.8	18.8	43.0
	% Within Age	0%	15.4%	22.7%	51.8%	33.6%
Total	Count	2	26	44	56	128
	Expected Count	2.0	26.0	44.0	56.0	128.0
	% Within Age	100.0%	100.0%	100.0%	100.0%	100.0%

*χ*^2^ = 15.513, *df* = 3, *p* = 0.001.

**Table 7 diseases-10-00068-t007:** Cross-tabulation results for research study focusing on asthma by age.

	Age					Total
		Up to 18	19–45	46–65	66 and Above	
Interested in participating in research focusing on asthma	Count	0	19	30	50	99
	Expected Count	1.5	20.1	34.0	43.3	99.0
	% Within Age	0%	73.1%	68.2%	89.3%	77.3%
Not interested in participating in research focusing on asthma	Count	0	19	30	50	99
	Expected Count	1.5	20.1	34.0	43.3	99.0
	% Within Age	0%	73.1%	68.2%	89.3%	77.3%
Total	Count	2	26	44	56	128
	Expected Count	2.0	26.0	44.0	56.0	128.0
	% Within Age	100.0%	100.0%	100.0%	100.0%	100.0%

*χ*^2^ = 13.763, *df* = 3, *p* = 0.003.

**Table 8 diseases-10-00068-t008:** Interest in Research Study Focusing on Mental Health by Age.

	Age					Total
		Up to 18	19–45	46–65	66 and Above	
Interested in participating in research focusing on mental health	Count	0	16	20	16	52
	Expected Count	0.8	10.6	17.9	22.8	52.0
	% Within Age	0%	61.5%	45.5%	28.6%	40.6%
Not interested in participating in research focusing on mental health	Count	2	10	24	40	76
	Expected Count	1.2	15.4	26.1	33.3	76.0
	% Within Age	100.0%	38.5%	54.5%	71.4%	59.4%
Total	Count	2	26	44	56	128
	Expected Count	2.0	26.0	44.0	56.0	128.0
	% Within Age	100.0%	100.0%	100.0%	100.0%	100.0%

*χ*^2^ = 9.881, *df* = 3, *p* = 0.020.

**Table 9 diseases-10-00068-t009:** Importance of Trust for Working with the JSU JHS CORC by Age.

	Age					Total
		Up to 18	19–45	46–65	66 and Above	
Not important	Count	0	2	0	1	3
	Expected Count	0.1	0.7	1.1	1.2	3.0
	% Within Age	0%	8.0%	0%	2.2%	2.7%
Slightly important	Count	0	1	1	1	3
	Expected Count	0.1	0.7	1.1	1.2	3.0
	% Within Age	0%	4.0%	2.5%	2.2%	2.7%
Neutral	Count	2	3	2	7	14
	Expected Count	0.2	3.1	5.0	5.7	14.0
	% Within Age	100.0%	12.0%	5.0%	15.2%	12.4%
Very important	Count	0	10	14	22	46
	Expected Count	0.8	10.2	16.3	18.7	46.0
	% Within Age	0%	36.0%	57.5%	32.6%	41.6%
Extremely important	Count	0	9	23	15	47
	Expected Count	0.8	10.4	16.6	19.1	47.0
	% Within Age	0%	36.0%	57.5%	32.6%	41.6%
Total	Count	2	25	40	46	113
	Expected Count	2.0	25.0	40.0	46.0	113.0
	% Within Age	100.0%	100.0%	100.0%	100.0%	100.0%

*χ*^2^ = 24.622, *df* = 12, *p* = 0.017.

**Table 10 diseases-10-00068-t010:** Importance of Trust While Working with the JSU JHS CORC by Sex.

	Sex			Total
		Male	Female	
Not important	Count	2	1	3
	Expected Count	0.7	2.3	3.0
	% Within Sex	8.0%	1.2%	2.8%
Slightly important	Count	0	3	3
	Expected Count	0.7	2.3	3.0
	% Within Sex	0%	3.6%	2.8%
Neutral	Count	3	11	14
	Expected Count	3.2	10.8	14.0
	% Within Sex	12.0%	13.3%	13.0%
Very important	Count	15	30	45
	Expected Count	10.4	34.6	45.0
	% Within Sex	60.0%	36.1%	41.7%
Extremely important	Count	5	38	43
	Expected Count	10.0	33.0	43.0
	% Within Sex	20.0%	45.8%	39.8%
Total	Count	25	83	108
	Expected Count	25.0	83.0	108.0
	% Within Sex	100.0%	100.0%	100.0%

*χ*^2^ = 9.953, *df* = 4, *p* = 0.041.

**Table 11 diseases-10-00068-t011:** Email as the Best Way to Communicate to Respondents About Community Outreach Activities by County of Residence.

	County			Total
		Hinds	Other Combined	
Email is the best way to communicate about community outreach activities	Count	33	27	60
	Expected Count	40.8	19.2	60.0
	% Within County	37.9%	65.9%	46.9%
Email is not the best way to communicate about community outreach activities	Count	54	14	68
	Expected Count	46.2	21.8	68.0
	% Within County	62.1%	34.1%	53.1%
Total	Count	87	41	128
	Expected Count	87.0	41.0	128.0
	% Within County	100.0%	100.0	100.0%

*χ*^2^ = 8.725, *df* = 1, *p* = 0.003.

**Table 12 diseases-10-00068-t012:** Text Messages as the Best Way to Communicate to Respondents about Community Outreach Activities by County of Residence.

	County			Total
		Hinds	Other	
Text messages is the best way to communicate about community outreach activities	Count	17	17	34
	Expected Count	23.1	10.9	34.0
	% Within County	19.5%	41.5%	26.6%
Text messages is not the best way to communicate about community outreach activities	Count	70	24	94
	Expected Count	63.9	30.1	94.0
	% Within County	80.5%	58.5%	73.4%
Total	Count	87	41	128
	Expected Count	87.0	41.0	128.0
	% Within County	100.0%	100.0	100.0%

*χ*^2^ = 6.866, *df* = 1, *p* = 0.009.

**Table 13 diseases-10-00068-t013:** Mail Reminders as the Best Way to Communicate to Respondents about Community Outreach Activities by Sex.

	Sex			Total
		Male	Female	
Mail reminders are the best way to communicate about community outreach activities	Count	8	50	58
	Expected Count	13.7	44.3	58.0
	% Within Sex	27.6%	53.2%	47.2%
Mail reminders are not the best way to communicate about community outreach activities	Count	21	44	65
	Expected Count	15.3	49.7	65.0
	% Within Sex	72.4%	46.8%	52.8%
Total	Count	29	94	123
	Expected Count	29.0	94.0	123.0
	% Within Sex	100.0%	100.0	100.0%

*χ*^2^ = 5.831, *df* = 1, *p* = 0.016.

**Table 14 diseases-10-00068-t014:** Cross-tabulation results for email being the best way to communicate to respondents about community outreach activities by age.

	Age					Total
		Up to 18	19–45	46–65	66 and Over	
Email is the best way to communicate about community outreach activities	Count	1	19	23	17	60
	Expected Count	0.9	12.2	20.6	26.3	60.0
	% Within Age	50.0%	73.1%	52.3%	30.4%	46.9%
Email is not the best way to communicate about community outreach activities	Count	1	7	21	39	68
	Expected Count	1.1	13.8	23.4	29.8	68.0
	% Within Age	50.0%	26.9%	47.7%	69.6%	53.1%
Total	Count	2	26	44	56	128
	Expected Count	2.0	26.0	44.0	56.0	128.0
	% Within Age	100.0%	100.0%	100.0%	100.0%	100.0%

*χ*^2^ = 13.826, *df* = 3, *p* = 0.003.

**Table 15 diseases-10-00068-t015:** Telephone Follow-Up Calls as the Best Way to Communicate to Respondents about Community Outreach Activities by Age.

	Age					Total
		Up to 18	19–45	46–65	66 and Over	
Telephone calls are the best way to communicate about community outreach activities	Count	0	4	18	31	53
	Expected Count	0.8	10.8	18.2	23.2	53.0
	% Within Age	0%	15.4%	40.9%	55.4%	41.4%
Telephone calls are not the best way to communicate about community outreach activities	Count	2	22	26	25	75
	Expected Count	1.2	15.2	25.8	32.8	75.0
	% Within Age	100.0%	84.6%	59.1%	44.6%	58.6%
Total	Count	2	26	44	56	128
	Expected Count	2.0	26.0	44.0	56.0	128.0
	% Within Age	100.0%	100.0%	100.0%	100.0%	100.0%

*χ*^2^ = 13.167, *df* = 3, *p* = 0.004.

**Table 16 diseases-10-00068-t016:** Text Messages as the Best Way to Communicate with Respondents about Community Outreach Activities by Age.

	Age					Total
		Up to 18	19–45	46–65	66 and Over	
Text messages are the best way to communicate about community outreach activities	Count	2	10	13	9	34
	Expected Count	0.5	6.9	11.7	14.9	34.0
	% Within County	100.0%	38.5%	29.5%	16.1%	26.6%
Text messages are not the best way to communicate about community outreach activities	Count	0	16	31	47	94
	Expected Count	1.5	19.1	32.3	41.1	94.0
	% Within County	0%	61.5%	70.5%	83.9%	73.4%
Total	Count	2	26	44	56	128
	Expected Count	2.0	26.0	44.0	56.0	128.0
	% Within County	100.0%	100.0%	100.0%	100.0%	100.0%

*χ*^2^ = 10.777, *df* = 3, *p* = 0.013.

**Table 17 diseases-10-00068-t017:** Mail Reminders as the Best Way to Communicate to Respondents about Community Outreach Activities by Age.

	Age					Total
		Up to 18	19–45	46–65	66 and Over	
Mail reminders are the best way to communicate about community outreach activities	Count	1	5	24	30	60
	Expected Count	0.9	12.2	20.6	26.3	60.0
	% Within County	50.0%	19.2%	54.5%	53.6%	46.9%
Mail reminders are not the best way to communicate about community outreach activities	Count	1	21	20	26	68
	Expected Count	1.1	13.8	23.4	29.8	68.0
	% Within County	50.0%	80.8%	45.5%	46.4%	53.1%
Total	Count	2	26	44	56	128
	Expected Count	2.0	26.0	44.0	56.0	128.0
	% Within County	100.0%	100.0%	100.0%	100.0%	100.0%

*χ*^2^ = 10.035, *df* = 3, *p* = 0.018.

**Table 18 diseases-10-00068-t018:** Perceptions of Heart Disease Affecting Community the Most by County of Residence.

	County			Total
		Hinds	Other Combined	
Heart disease affecting community the most	Count	35	9	44
	Expected Count	29.9	14.1	44.0
	% Within County	40.2%	22.0%	34.4%
Heart disease not affecting community the most	Count	52	32	84
	Expected Count	57.1	26.9	84.0
	% Within County	59.8%	78.0%	65.6%
Total	Count	87	41	128
	Expected Count	87.0	41.0	128.0
	% Within County	100.0%	100.0	100.0%

*χ*^2^ = 4.127, *df* = 1, *p* = 0.042.

**Table 19 diseases-10-00068-t019:** High Blood Pressure Affecting Community the Most by Sex.

	Sex			Total
		Male	Female	
High Blood Pressure affecting community the most	Count	10	12	22
	Expected Count	5.2	16.8	22
	% Within Sex	34.5%	12.8%	17.9%
High Blood Pressure not affecting community the most	Count	19	82	101
	Expected Count	23.8	77.2	101.0
	% Within Sex	65.5%	87.2%	82.1%
Total	Count	29	94	123
	Expected Count	29.0	94.0	123.0
	% Within Sex	100.0%	100.0	100.0%

*χ*^2^ = 4.127, *df* = 1, *p* = 0.042.

## Data Availability

Not applicable.

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
