# Peer review of "Evaluating Levels of Community Participation in a University-Community Partnership: The Jackson Heart Study"

_diseases, 2022, doi:10.3390/diseases10040068_

Round 1

Reviewer 1 Report

Dear colleagues

I am always delighted to see CBPR studies and applaud you for your efforts. In terms of presentation in this paper, my main comment is that I found it hard to get from the methods whether you are presenting findings from the focus groups or the survey here, and I was confused by the presentation in counts/expected counts. Surely it couldn't have been the 128 participants and then counts like "3" are supposed to have been how many answered? Please clarify this in the methods section and also add a table with descriptive characteristics of the sample that you analysed here. 

Please add a section on limitations of your research in light of the participatory aspects: how indeed were the participants involved in this particular survey and analysis?

Author Response

(Reviewer 1)

Reviewer’s Comments:    my main comment is that I found it hard to get from the methods whether you are presenting findings from the focus groups or the survey here, and I was confused by the presentation in counts/expected counts. Surely it couldn't have been the 128 participants and then counts like "3" are supposed to have been how many answered? Please clarify this in the methods section and also add a table with descriptive characteristics of the sample that you analysed here. 

Authors’ Response #1: For each of the questions, the participants were asked to select their response choice. The corresponding Chi-Square contingency table enumerates the frequency of each choice selected by the participants. The numbers like the “3” that was mentioned represents the number of participants who selected that particular response with the total number of participants (128) recording choices listed as “TOTAL” at the bottom of the column, depending upon whether all participants provided responses for all questions.

The following Section has been added to the Methods Section for clarity in response to the suggestion: All of the attendees at the event were invited to participate in the survey. We analyzed data from 128 of the participants who provided completed surveys. The participants’ responses to each area of interest on the survey are provided in the Chi-Square tables in the results section with a tabulation of the frequency of selection of each of the Likert scale choices among the 128 participants (enumerating the number of participants who selected Strongly Agree, Agree, Neutral, Disagree, and Strongly Disagree).

Authors’ Response #2: A table with the descriptive characteristics of the participants is included as suggested.

Reviewer’s Comment: Please add a section on limitations of your research in light of the participatory aspects: how indeed were the participants involved in this particular survey and analysis?

Authors’ Response #3: The following section on “Limitations and Strengths” has been added as recommended.

Limitations and Strengths

This study has both limitations and strengths. Using a convenience sampling strategy in selecting the study participants is a limitation because some important JHS participants and community partners might have been missed. The small number of participants is a limitation. Some bias may be present because the research was conducted at a JHS community event. As a result, participants would have had a heightened awareness of the communication being administered to them because they were themselves, discussing how to effectively disseminate messages to their community audiences.  

The design of this study provided numerous benefits. The choice to administer the survey to a group that included JHS cohort members, family members, local community, and other stakeholders around the Jackson Metropolitan Area, was also a benefit as it meant data were based on a variety of first-hand experiences from individuals with an active interest in promoting JHS activities. The results are therefore applicable and meaningful to CBPR research. The limitations mentioned suggest that our findings might not be easily generalizable to the population. Therefore, additional studies are encouraged to investigate whether different populations’ perceptions of appropriate communication strategies are different from our findings. The lessons learned from this study should be received as meaningful resources when developing strategies to inform the development of specific health information dissemination solutions.

Reviewer 2 Report

The original research manuscript entitled "Evaluating Levels of Community Participation in a University-Community Partnership: The Jackson Heart Study" is a very interesting paper that reports aspects of the relationship between community and medical facilities and projects. Some minor aspects should be reviewed:

> Was the sample size calculated so that the results could achive statitstical power? If this is not the case a limitations paragraph should be added

> The sample was elected randomly? If this is the case, the randomization method should be explained. If this is not the case, also a reference to this issue should be described in a limitations paragraph

> Questions in the results section should be moved to methods section. A supplementary section with the questionnaire should be added

Author Response

(Reviewer 2)

Reviewer’s Comments: The original research manuscript entitled "Evaluating Levels of Community Participation in a University-Community Partnership: The Jackson Heart Study" is a very interesting paper that reports aspects of the relationship between community and medical facilities and projects. Some minor aspects should be reviewed:

> Was the sample size calculated so that the results could achive statitstical power? If this is not the case a limitations paragraph should be added

> The sample was elected randomly? If this is the case, the randomization method should be explained. If this is not the case, also a reference to this issue should be described in a limitations paragraph

> Questions in the results section should be moved to methods section. A supplementary section with the questionnaire should be added

Authors’ Response #1: The following section on “Limitations and Strengths” has been added to address the reviewer’s concerns:

Limitations and Strengths

This study has both limitations and strengths. Using a convenience sampling strategy in selecting the study participants is a limitation because some important JHS participants and community partners might have been missed. The small number of participants is a limitation. Some bias may be present because the research was conducted at a JHS community event. As a result, participants would have had a heightened awareness of the communication being administered to them because they were themselves, discussing how to effectively disseminate messages to their community audiences.  

The design of this study provided numerous benefits. The choice to administer the survey to a group that included JHS cohort members, family members, local community, and other stakeholders around the Jackson Metropolitan Area, was also a benefit as it meant data were based on a variety of first-hand experiences from individuals with an active interest in promoting JHS activities. The results are therefore applicable and meaningful to CBPR research. The limitations mentioned suggest that our findings might not be easily generalizable to the population. Therefore, additional studies are encouraged to investigate whether different populations’ perceptions of appropriate communication strategies are different from our findings. The lessons learned from this study should be received as meaningful resources when developing strategies to inform the development of specific health information dissemination solutions.

Authors’ Response #2: The following questions were removed from the results section and added to the Methods Section, as recommended.

Community participants were asked to provide answers to the following questions:

  1. What are some ways we can increase participation in the community outreach activities?
  2. To what extent did the following reasons cause you to first participate in the community outreach activities?
  3. Would you be interested in participating in a community research study similar to the Jackson Heart Study?
  4. How important are the following items in keeping you engaged in the community outreach activities?
  5. What is the best way to communicate to you about community outreach activities?
  6. What diseases affect the community where you live the most?

Authors’ Response #3: We decided not to add the entire questionnaire in a supplementary section because the entire questionnaire was not used for this study. This study only examined six selected questions (included in the Methods Section) that reflected the focus of this research.

Round 2

Reviewer 1 Report

Thanks for addressing my comments